# A thiol probe for measuring unfolded protein load and proteostasis in cells

Moore Z. Chen[1], Nagaraj S. Moily[1], Jessica L. Bridgford[1], Rebecca J. Wood[1], Mona Radwan[1], Trevor A. Smith [2], Zhegang Song[3], Ben Zhong Tang [3], Leann Tilley[1], Xiaohong Xu[4], Gavin E. Reid[1,2], Mahmoud A. Pouladi [4,5], Yuning Hong [1,2,6] & Danny M. Hatters [1]

When proteostasis becomes unbalanced, unfolded proteins can accumulate and aggregate. Here we report that the dye, tetraphenylethene maleimide (TPE-MI) can be used to measure cellular unfolded protein load. TPE-MI fluorescence is activated upon labelling free cysteine thiols, normally buried in the core of globular proteins that are exposed upon unfolding. Crucially TPE-MI does not become fluorescent when conjugated to soluble glutathione. We find that TPE-MI fluorescence is enhanced upon reaction with cellular proteomes under conditions promoting accumulation of unfolded proteins. TPE-MI reactivity can be used to track which proteins expose more cysteine residues under stress through proteomic analysis. We show that TPE-MI can report imbalances in proteostasis in induced pluripotent stem cell models of Huntington disease, as well as cells transfected with mutant Huntington exon 1 before the formation of visible aggregates. TPE-MI also detects protein damage following dihydroartemisinin treatment of the malaria parasites *Plasmodium falciparum*. TPE-MI therefore holds promise as a tool to probe proteostasis mechanisms in disease.

---

[1] Department of Biochemistry and Molecular Biology and Bio21 Molecular Science and Biotechnology Institute, The University of Melbourne, Parkville, VIC 3010, Australia. [2] School of Chemistry, The University of Melbourne, Parkville, VIC 3010, Australia. [3] Department of Chemistry, The Hong Kong University of Science & Technology, Clear Water Bay, Kowloon, Hong Kong, China. [4] Translational Laboratory in Genetic Medicine, Agency for Science, Technology and Research, Singapore (A*STAR), 8A Biomedical Grove, Immunos, Level 5, Singapore 138648, Singapore. [5] Department of Medicine, Yong Loo Lin School of Medicine, National University of Singapore, Singapore 117597, Singapore. [6] Department of Chemistry and Physics, La Trobe Institute for Molecular Science, La Trobe University, Melbourne, VIC 3083, Australia. Correspondence and requests for materials should be addressed to Y.H. (email: Y.Hong@latrobe.edu.au) or to D.M.H. (email: dhatters@unimelb.edu.au)

The synthesis of new protein chains and their subsequent folding and targeting to specific cellular locations is one of the core activities of a cell. Because proteins typically fold inefficiently or display high inherent tendencies to misfold and aggregate, the folding process is strictly guided by quality control machinery[1]. Small globular proteins may fold and unfold reversibly and spontaneously[2]. However, complex proteins usually do not and this includes about two-thirds of the proteome[3]. In particular, proteins that fold in the endoplasmic reticulum (ER) need quality control assistance, including being directed through quality control checkpoints in the calnexin cycle[3, 4].

The quality control process and collective management of protein folding and turnover is defined as 'proteostasis', and involves a network of over 500 proteins in humans (as well as conserved systems in all life forms including bacteria and plants)[5]. When proteome foldedness is challenged through acute stresses, mechanisms are activated to improve baseline proteostasis capacity and to rectify imbalances. One example of this is the heat shock response whereby heat shock protein family chaperones are upregulated to help restore proteome foldedness[6]. Another example is the unfolded protein response which targets blocks in the folding pipeline of ER proteins[7]. Collectively these and other proteostasis mechanisms can buffer against short term strains on proteome folding.

In many diseases, especially neurodegeneration, proteostasis can become chronically imbalanced. Markers of this imbalance include expression of ER stress genes, upregulation of autophagy and inappropriate protein aggregation[7–9]. Indeed, chronic imbalance can be induced in animal models by ectopic expression of unstably folded proteins, which overwhelms the finite capacity of quality control reserves[10, 11].

One outstanding challenge in the field is determining the baseline efficiency with which cells maintain proteostasis. To date, the best approach to understand this is by expression of meta-stable proteins prone to aggregation, where the extent of aggregation is used as a proxy for proteostasis[12, 13]. Here we describe the development of a strategy using a simple fluorogenic dye that can capture a snapshot of the balance of unfolded protein relative to folded states and does not require any ectopic expression of protein reporters. With purified proteins, and cellular models of proteome folding stress we validate this strategy and show three examples of how tetraphenylethene maleimide (TPE-MI) can measure proteostasis imbalance in disease contexts.

## Results

### TPE-MI is a turn-on fluorescent probe of unfolded proteins.
We posited that under folding stresses cells will accumulate a backlog of unfolded proteins, consisting of proteins requiring quality control to fold and/or already-folded proteins that are subjected to en masse unfolding. Hence, our strategy to monitor proteostasis efficiency was to label unfolded proteome. Non-disulphide bonded cysteine (i.e., with free thiols) is the least surface-exposed residue of all amino acids in proteins[14]. Thus, the abundance of accessible thiols should increase in cells as the proteome becomes more unfolded, enabling us to target these thiols as a proxy reporter for the level of unfolded proteins (Fig. 1a). Similar strategies have been used to study the protein folding state of purified globular proteins in vitro by measuring the loss of reactivity of free cysteines as they become buried in the core of proteins in the folded state, using the colorimetric Ellman's reagent (5,5′-dithio-bis-[2-nitrobenzoic acid])[15]. In cells, a key requirement for such a strategy is to use a dye that not only permeates the cell membrane but becomes fluorescent only once it has reacted with the target cysteines.

MI-functionalized TPE is a fluorescent dye with attributes that we predicted would make it suitable for this purpose (Fig. 1b). First, TPE-MI is inherently non-fluorescent until it is conjugated to a thiol via the maleimide[16]. This lack of fluorescence has been postulated to be due to the MI group quenching the TPE emission through the $n-\pi$ electronic conjugation of the carbonyl and olefinic groups, which are disrupted by thiol conjugation as evidenced by the blue-shift of the absorption maximum (Supplementary Fig. 1). TPE fluorescence is also contingent on its aggregation-induced emission (AIE) property whereby the fluorescence quantum yield is dependent on the motional restriction of the four phenyl rotamers of the TPE fluorophore that occurs in a rigid local molecular environment[17].

To determine whether TPE-MI could be used to probe protein unfolding, we first investigated its reactivity to several model proteins: three that contain single free cysteines buried in the core of the folded state (bovine β-lactoglobulin, yeast enolase and human peroxiredoxin-3) and one negative control (for specificity of the reaction) that lacks cysteine (ubiquitin). Each of the proteins that contain a buried free cysteine thiol displayed far greater reactivity to TPE-MI when unfolded with guanidine hydrochloride, consistent with TPE-MI being selectively reactive with thiols that become exposed when globular proteins are unfolded (Fig. 2a–c). Ubiquitin showed no reactivity with TPE-MI, which demonstrated the specificity of the dye for thiols (Fig. 2a). Specificity for cysteine was further demonstrated by adding the competitive thiol reactant, N-methylmaleimide (NMM), in combination with TPE-MI to denatured β-lactoglobulin, which resulted in no increase in TPE fluorescence (Fig. 2d). TPE-MI alone, or protein alone also showed no changes in fluorescence signal (Supplementary Fig. 2).

Inside cells, glutathione (GSH) represents a major pool of non-protein thiols. It is present at concentrations of ~5 mM[18],

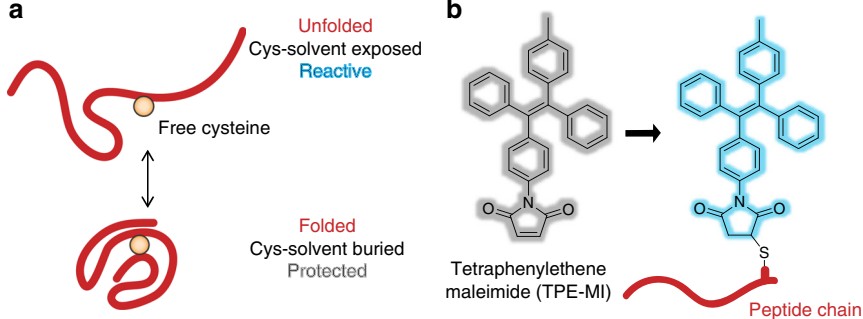

**Fig. 1** Strategy for assaying protein foldedness via access to buried cysteine thiols. **a** Strategy for probing free cysteine thiols that become exposed to the solvent upon protein unfolding and permissive to maleimide reaction. **b** Structure of tetraphenylethene (TPE) conjugated to a maleimide (MI). Fluorescence is enabled upon conjugation to a protein and immobilisation of the phenyl rotamers.

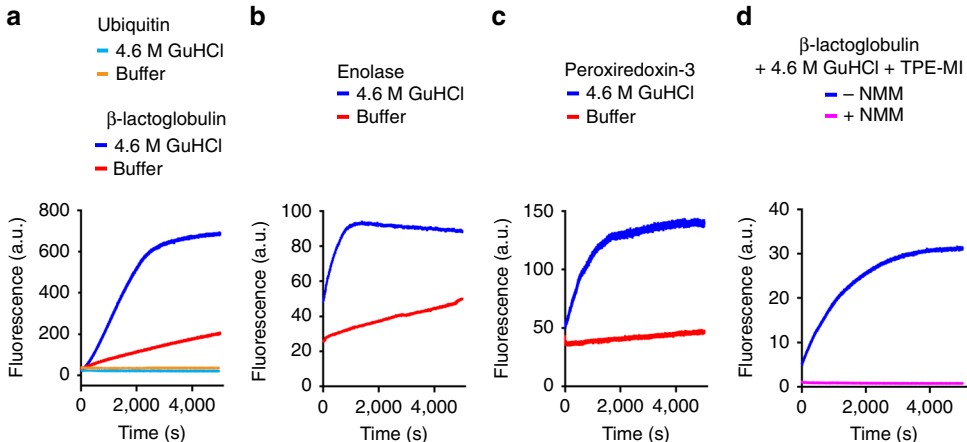

**Fig. 2** TPE-MI preferentially reacts with buried cysteine thiols in unfolded proteins and switches on fluorescence. Shown are representative reaction kinetics for four proteins (note that the absolute fluorescence values between graphs here and in other figures cannot be compared to each other due to differences in instrument settings). **a** Bovine β-lactoglobulin, which contains five thiols: one buried free thiol and four disulphide-linked thiols; and bovine ubiquitin, which contains no thiols. **b** *Saccharomyces cerevisiae* enolase, which contains one buried free thiol and **c** human peroxiredoxin 3, which contains three thiols: one buried free thiol and two disulphide-linked surface-exposed thiols. Proteins were suspended in 100 mM sodium phosphate, pH 7.4, alone or supplemented with guandine hydrochloride (GuHCl) to induce denaturation. At the start of the reaction 50 μM TPE-MI was added. **d** Same design but with 50 μM *N*-methylmaleimide (NMM) added before the addition of TPE-MI.

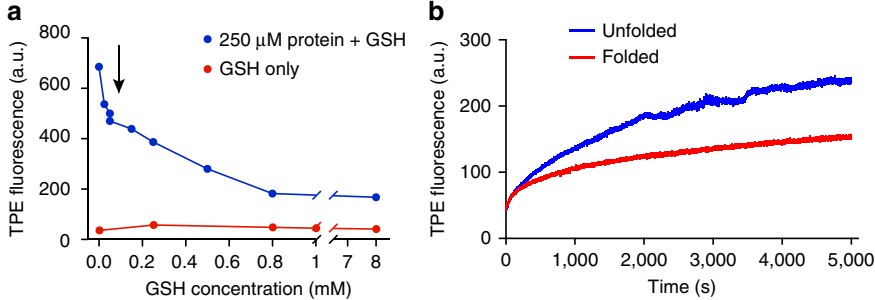

**Fig. 3** TPE-MI fluorescence is not activated by glutathione reaction and can detect increases in unfolded protein load in cell lysates. **a** Reactivity of TPE-MI (50 μM) with β-lactoglobulin (250 μM) in the presence of the small thiol containing peptide glutathione (0.025 mM–8 mM GSH) and 4.6 M GuHCl. Fluorescence intensity of TPE-MI with different concentrations of GSH is shown for comparison (measured in 4.6 M GuHCl). Arrow indicates 7:3 protein thiol to glutathione thiol ratio found intracellularly. **b** Reaction kinetics of 50 μM TPE-MI in the presence of intact and unfolded cell lysates from mouse Neuro2a neuroblastoma cells (0.5 mg ml⁻¹ total cellular protein). Shown are 'native' cell lysate in 100 mM sodium phosphate, pH 7.4 and 'unfolded' cell lysate containing 4.6 M GuHCl.

representing a GSH thiol to protein thiol molar ratio of approximately 3:7[19]. Hence GSH provides a major potential competitor that could interfere with the application of TPE-MI to probe the unfolded protein load. However, we discovered that GSH-conjugated TPE-MI is non-fluorescent (Fig. 3a and Supplementary Fig. 3a) (Supplementary Fig. 4a, b) confirmed the successful conjugation of GSH to TPE-MI). This contrasts with a previous report that TPE-MI conjugated to GSH is highly fluorescent[16]. However, in that case the GSH-TPE-MI conjugate was assessed in the solid phase[16]. Hence, it appears that in aqueous solution the phenyl rotamers of TPE-MI remain mobile when conjugated to the small GSH tripeptide. In contrast, it appears that proteins provide sufficient rigidity to enable AIE. This may arise from localised clustering of the phenyl rotamers with nearby hydrophobic amino acid sidechains that typically comprise the core of globular proteins. Support for this mechanism was revealed by time-resolved fluorescence measurements, which showed that TPE-MI emission decayed much faster when conjugated to GSH compared to the unfolded protein conjugate (Supplementary Fig. 5 and Supplementary Table 1). Collectively, these results suggested that in the cellular context GSH will have only a small dampening effect on the fluorescence

yields arising from TPE-protein thiol conjugates. Indeed the fluorescence yield of β-lactoglobulin remained high in the presence of glutathione at relevant cellular ratios of proteins and glutathione thiols (Fig. 3a). Further supporting this conclusion is the observation that denaturation of proteins in whole cell lysate with guanidine hydrochloride increased the reactivity with TPE-MI despite the presence of GSH (Fig. 3b).

**TPE-MI detection of altered unfolded protein load in cells.** Next we assessed the behaviour of TPE-MI in live cells. TPE-MI exhibited a homogeneous cytoplasmic labelling pattern in live HeLa cells, with a lower level of labelling in the nucleus and apparent concentration in the region of the ER, which was anticipated as a major location for protein synthesis and folding (Fig. 4a). This indicated a good permeation of the plasma membrane and accessibility of the reagent to the intracellular proteome. Pre-treatment of the cells with NMM, which is known to permeate the cell membrane and block intracellular thiols[20–22], led to a reduction in labelling, suggesting that the TPE-MI fluorescence is associated with specific modification of thiol groups in the cell (Supplementary Fig. 6). TPE-MI was not

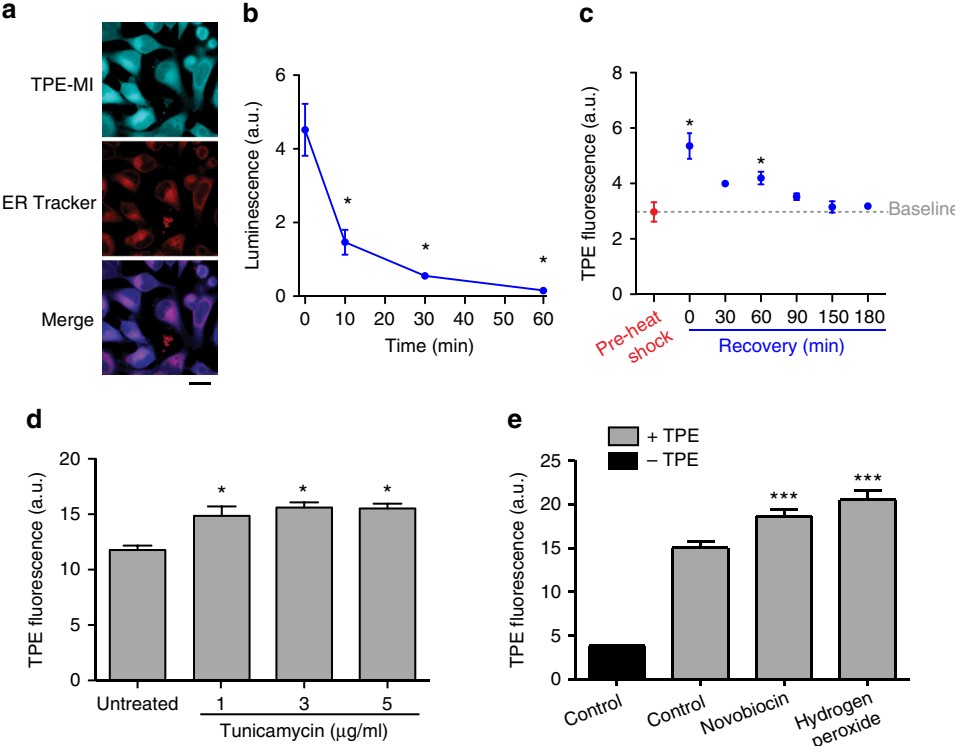

**Fig. 4** TPE-MI fluorescence increases in accordance with diverse proteome stresses. **a** Confocal microscopy images for TPE-MI staining of HeLa cells with ER Tracker counterstain. Cells were fixed in 4% (v/v) paraformaldehyde for 15 min after staining with 50 μM TPE-MI. Scale bar, 20 μm. **b** Unfolding of the proteome by heat shock as assessed by denaturation of *Renilla* luciferase and luciferase activity. Data show luciferase luminescence of lysates taken after heat shock at 42 °C of HeLa cells transfected with *Renilla* luciferase for the times shown, n = 5; mean ± s.e.m. **c** Heat shock treatment of HeLa cells at 42 °C for 45 min and subsequent time course of recovery at 37 °C. Data show median TPE-MI fluorescence measured by flow cytometry of cells stained at the indicated time points, n = 5 technical replicates; mean ± s.e.m. Statistics for **b**, **c** were calculated between test groups and basal groups or zero time point. The details of the statistics are provided in Supplementary Data 1 where P < 0.05 (indicated by *). **d** Effect of overnight tunicamycin treatment on HeLa cells. Data show median fluorescence intensity of TPE-MI stained cells measured by flow cytometry, n = 5 biological replicates; mean ± s.e.m. **e** Effects of hsp90 inhibitor novobiocin (800 μM for 6 h) and free radical generator, hydrogen peroxide (100 μM for 1 h) on HEK293 cells. Data show means ± s.d. for five replicates. ***P < 0.0001. See Supplementary Data 1 for full statistics.

noticeably toxic to cells at concentrations and time points used in our assays (or at higher concentrations) as measured by death rates (Supplementary Fig. 7).

To test how effective TPE-MI was in monitoring unfolded protein load in live cells, we applied a variety of distinct stresses to proteome foldedness. First, we applied a 42 °C heat shock pulse to thermally unfold the proteome[23]. As anticipated, the heat shock led to an increase in non-functional and unfolded proteins within ~30 min, as measured by a luciferase activity reporter assay (Fig. 4b; full details of statistical tests in Supplementary Data 1). TPE-MI treatment (on aliquots of cells) revealed an immediate increase in reactivity after heat shock. However, upon recovery at 37 °C, the levels of reactivity (on aliquots of cells) returned to baseline levels within about 150 min, which is in accord with the anticipated time for HeLa cells to recover from heat shock[24] (Fig. 4c). This result indicated that TPE-MI can measure net increases in the unfolded protein load and recovery from thermal denaturation. Other stresses also increased TPE fluorescence in accord with an anticipated accumulation of unfolded protein (Fig. 4d, e). These included tunicamycin, which is a glycosylation inhibitor and leads to a backlog of unfolded protein in the ER that cannot complete the calnexin cycle[25], inhibition of hub chaperone Hsp90 with novobiocin, which impairs Hsp90 activity without activating the heat shock response[26, 27] and hydrogen peroxide, which confers damage by free radical reactions.

Next we tested whether TPE-MI could be used to track foldedness of individual proteins by proteomics approaches.

For these experiments we used tunicamycin treatment as the folding stress. Sodium dodecyl sulphate–polyacrylamide gel electrophoresis (SDS–PAGE) profiles of cell lysates revealed tunicamycin to induce increased TPE reactivity of a broad range of proteins, including a prominent protein at ~50 kDa (Fig. 5a, b). As a control for specificity, we investigated a non-alkylating analogue of TPE-MI, which did not stain either purified proteins with a free cysteine thiol, or change fluorescence in cells treated with tunicamycin (Supplementary Fig. 8a–c). For deeper insight we next performed a quantitative proteomics analysis using a matched pair design with the method of stable isotope labelling with amino acids in cell culture (SILAC)[28]. For this approach, we labelled control and tunicamycin-treated cells with TPE-MI (as the matched pairs) and then assessed the relative loss of peptides containing unmodified Cys residues from the mass spectrum. By performing nano-liquid chromatography-tandem mass spectrometry (LC-MS/MS), which was sufficient to sample a selection of the most abundant proteins of the proteome, we identified 154 Cys-containing proteins in all five replicates (Supplementary Data 2 for full proteomics summary). The median ratio of Cys-containing peptides in the tunicamycin/control conditions was significantly lower (by ~6%) than the ratio of non-Cys-containing peptides, which is consistent with tunicamycin treatment leading to a greater backlog of unfolded proteins (Fig. 6a).

Analysis of the individual Cys-containing peptides revealed a number of proteins that exhibited very large increases in

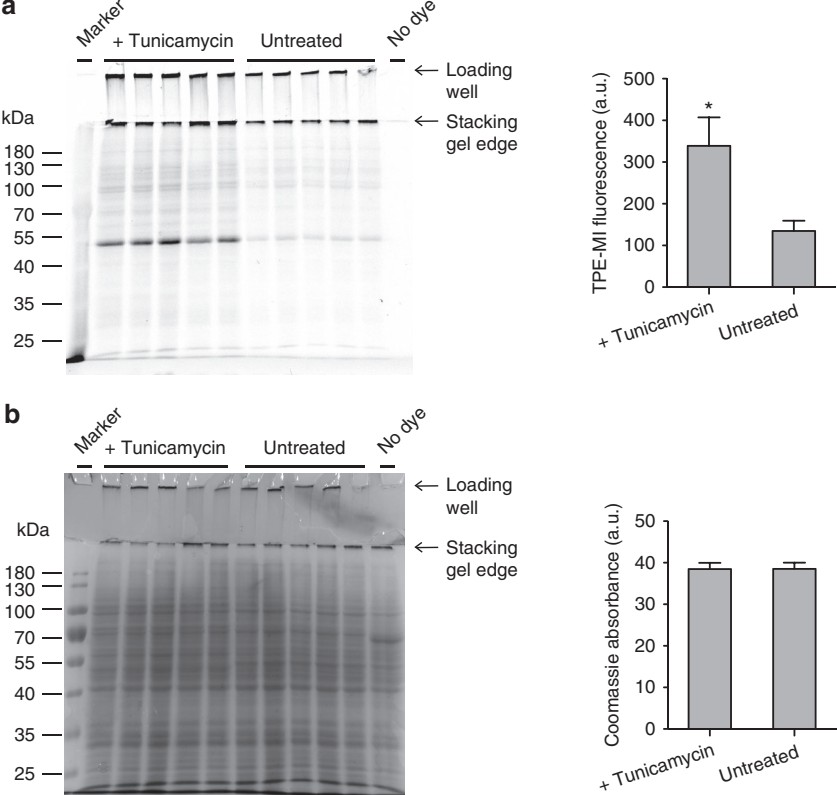

**Fig. 5** Detection of proteome-wide Cys-reactivity upon tunicamycin treatment. **a** In-gel fluorescence of TPE-MI stained Neuro2a cell lysates by SDS–PAGE. Cells were treated with tunicamycin (1 µg ml$^{-1}$, overnight) or vehicle control, then labelled with 50 µM TPE-MI for 30 min before lysis. 40 µg total cellular protein was loaded on the gel. Graphs show quantitation of fluorescence intensities in each lane ($n = 5$ biological replicates; mean ± s.e.m.; *, $P < 0.05$—see Supplementary Data 1 for full statistics). **b** Corresponding Coomassie stained gel of **a**, and quantification. Graph shows quantitation of the Commassie intensities ($n = 5$; mean ± s.e.m.; see Supplementary Data 1 for full statistics).

reactivity upon TPE-MI treatment following tunicamycin stress (Supplementary Fig. 9 and Fig. 6b). The six peptides with the greatest increase in reactivity came from THRB (prothrombin), G3P (glyceraldehyde-3-phosphate dehydrogenase (GAPDH)), ACTG (actin, cytoplasmic 2), FHL1 (four and a half LIM domains protein 1), FBRL (rRNA 2′-O-methyltransferase fibrillarin) and ENOB (β-enolase). These proteins have diverse and unrelated functions and none have known roles in thiol exchange mechanisms. The prominent protein observed by SDS–PAGE at ~50 kDa may be ENOB, which has a molecular mass of 47 kDa and is highly abundant in cells (in the top 5% of protein abundance based on the PaxDb Protein Abundance database[29]). Of note, ACTG contains four Cys-containing peptides that were detected in the data set, but only one of these (peptide 1–19) displayed a major increase in reactivity (Fig. 6c). Most noteworthy a very similar peptide (2–19), differing only by its lack of the amino terminal methionine, showed no change in TPE reactivity. Since N-terminal methionines are routinely removed from most proteins after synthesis[30], these data suggest that TPE-MI selectively labels newly synthesised ACTG upon tunicamycin treatment, which presumably is incompetent to fold.

One cysteine residue in a peptide from ERF1 (residues 333–344) was notably protected from TPE reactivity under tunicamycin stress. ERF1 (eukaryotic release factor 1) is involved in releasing newly synthesised peptides from the ribosome during translation[31]. ERF1 was also fourfold less abundant (as measured from the non-Cys-containing peptide ratios) following tunicamycin treatment, and in light of lower levels of ERF1 reducing translation rates[32], this would appear to be a protective response

to suppress protein synthesis. As part of its function, ERF1 forms a complex with cofactor ERF3 via a domain (residues 270–437) that encompasses the protected cysteine residue[33, 34]. Hence, the reduction in TPE-MI reactivity may indicate a conformational change or binding event associated with a change in the function of ERF1 to reduce protein expression under stress.

**Soluble mutant Huntingtin increases the unfolded proteome.** We examined whether TPE-MI could report on the impact of mutant protein expression on the global unfolded protein load in cells. Prior work has suggested that the expression of conformationally destabilised and/or aggregation-prone mutant proteins associated with neurodegenerative diseases can lead to a destabilization of proteostasis[11, 12, 35]. For this experiment we examined the effect of Huntingtin, which when mutated to contain a polyglutamine sequence that is expanded beyond 36 contiguous residues, causes Huntington disease (HD)[36]. We expressed the exon 1 fragment (Httex1), which is sufficient to invoke HD-like disease in mouse models[37], as a fusion to mCherry. At the high expression level, the mutant 97Q form of Httex1 was associated with an elevated TPE-MI fluorescence signal relative to a non-disease-causing 25Q form of Httex1 (Fig. 7a). TPE-MI did not directly label Httex1 (at least the aggregates), which does not contain any cysteine and hence the changes observed in TPE-MI reactivity are not due to direct labelling of the ectopic protein (Supplementary Fig. 10). This result suggests that mutant proteins associated with neurodegenerative disease can strain proteostasis capacity and lead to a backlog of unfolded proteins in the cell.

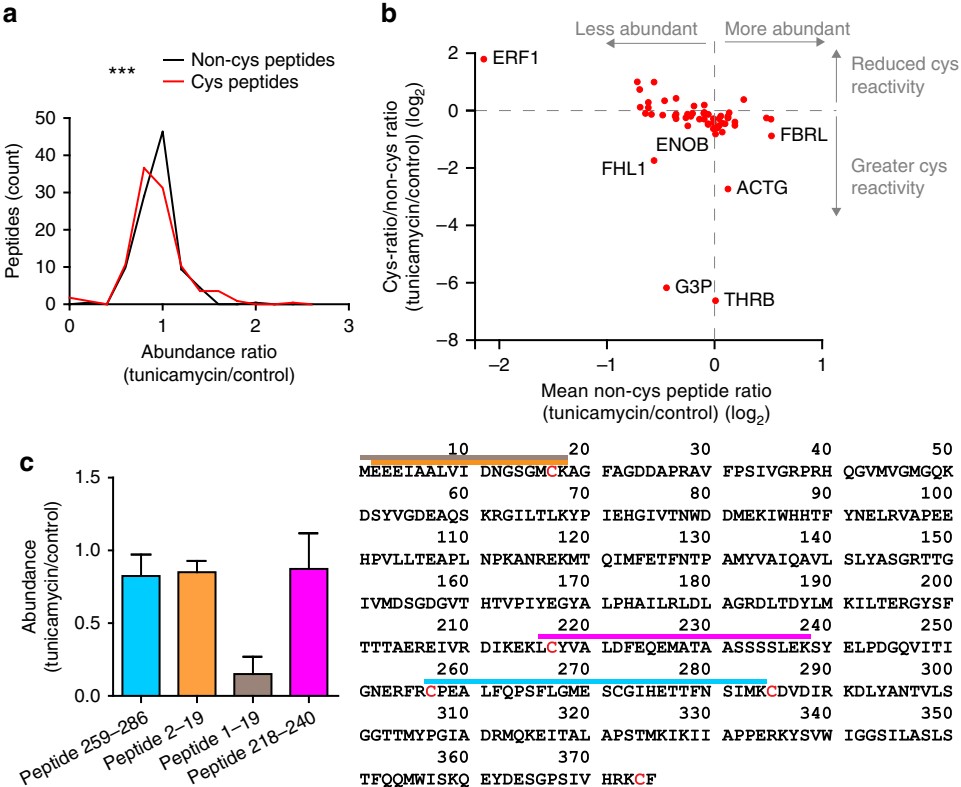

**Fig. 6** Tunicamycin-mediated block of folding selectively alters cysteine reactivity to TPE-MI. Data relates to Neuro-2a cells treated with TPE-MI (30 min; 200 μM) after overnight tunicamycin pre-treatment (1 μM) versus an untreated control. **a** Histogram of the peptide abundance ratios between tunicamycin-treated cells and control cells differentially labelled using SILAC. ***$P < 0.0001$ between two groups assessed by Wilcoxon matched-pairs signed rank test. See Supplementary Data 1 for full statistics. Differences indicate a tunicamycin-mediated loss of Cys residue-containing peptides attributable to increased TPE reaction. **b** Plot of peptide abundances (calculated from non-Cys peptide ratios) versus change of Cys reactivity (ratios of Cys peptides normalised to abundance by the non-cys peptides within the same protein) attributable to tunicamycin treatment. Shown are only peptides that display significant changes in Cys reactivity due to tunicamycin treatment ($P < 0.05$ measured by Student's $t$-test; details in Supplementary Data 2 and Supplementary Fig. 9). **c** Case study of peptides derived from ACTG. Shown are the mean ± s.d. peptide abundance ratios of five replicates (*left*) versus where the peptides map on the sequence of ACTG (*right*). Colours on bar graph are coordinated to the sequence map and cysteine residues are shown in red on the sequence.

The formation of visible protein aggregates that is associated with neurodegenerative disease has been previously suggested to reflect an adaptive, sequestration response to the stress exerted by mutant proteins[38–40]. Our TPE-MI probe provides a unique tool to explore the relationship between protein aggregation and the backlog of unfolded (or misfolded) protein in the cell. To investigate this relationship, we applied our previously developed flow cytometry approach, Pulse Shape Analysis (PulSA), to separate cells with visible Httex1 aggregates (inclusion (i) population) from cells with unaggregated Httex1 protein (non-inclusion (ni) population)[41]. Classification of the cells into the ni and i groups revealed that, at the lower levels of expression, cells without visible inclusions exhibited with a much higher TPE-MI signal than cells with inclusions (Fig. 7b). Conversely the cells with 97Q Httex1 aggregates had a TPE-MI signal below even the 25Q counterpart at lower levels of expression. Only at the highest levels of expression did cells with visible inclusions become highly elevated in TPE-MI fluorescence to levels similar to cells lacking aggregates. Collectively, these data are consistent with the Httex1 monomer placing the greatest strain on proteostasis leading to the rest of the proteome becoming backlogged with unfolded protein. In contrast, the very low level of TPE-MI fluorescence after an inclusion has formed suggested that inclusion formation removes the backlog of unfolded protein load.

To further explore the role that mutant Huntingtin has in proteostasis imbalance, we examined TPE-MI reactivity of induced pluripotent stem cells derived from HD patients differentiated into primitive neural stem cells (pNSCs)[42]. A HD cell line encoding 180Q in the endogenous Huntingtin protein (ND36999) showed greater induction of TPE-MI reactivity than a 'wild-type' cell line with 33Q (ND36997) upon stressing proteostasis in the ER and autophagy pathways with thapsigargin (Fig. 7c). Because these two lines come from genetically different backgrounds, we also tested TPE-MI reactivity in the 180Q line gene-edited back to a 'wild-type' polyQ length (18Q) by CRISPR/Cas9 and homologous recombination[42]. Three independently corrected lines all showed reduced induction of TPE reactivity when challenged with thapsigargan. These results all suggest HD cells have a reduced ability to prevent the build-up of unfolded proteins under stress and provide a further demonstration of how TPE-MI can probe the breakdown of proteostasis dynamics in disease.

**DHA promotes proteome unfolding in malaria parasites.** Our last experiment was to test TPE-MI as a probe for proteome destabilization in a completely different biological setting. Dihydroartemisinin (DHA) is a frontline drug for the treatment of malaria[43]. DHA has been postulated to be toxic to malaria

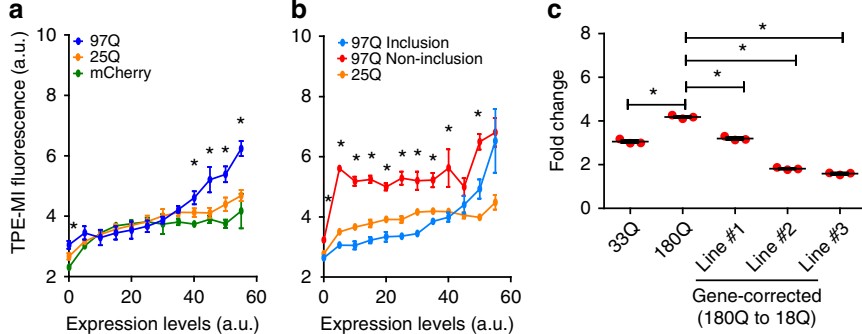

**Fig. 7** Backlog of unfolded protein in Neuro-2a cells triggered by mutant Huntingtin protein. **a** Impact of Htt exon 1 (Httex1) expression on TPE-MI fluorescence comparing the wild-type polyQ length (25Q) with a severe mutant (97Q) as fusions to mCherry. Cells were transfected and analysed by flow cytometry with the data binned into different expression levels based on mCherry fluorescence. Data show $n = 3$ technical replicates; mean ± s.e.m. *represents significant differences between 25QHttex1 and 97QHttex1 ($P < 0.05$). **b** Reanalysis of the data in panel a by PulSA to separate the cells with visible Httex1 inclusions from those without. Data show $n = 3$ technical replicates; mean ± s.e.m. *represents significant differences between 97QHttex1 inclusion and 97QHttex1 non-inclusion populations. **c** TPE reports on stress in primitive neural stem cells derived from induced pluripotent stem cells from Huntington patients. Shown are Huntington diease human neural cell lines encoding variable polyQ lengths in the *Huntingtin* gene. The gene corrected lines were gene-edited by CRISPR-Cas9 technology. Cells were analysed by flow cytometry and show fold change in ER stress induced by 1 μM thapsigargin relative to vehicle (dimethyl sulfoxide). Data represent $n = 3$ technical replicates, mean ± s.e.m. *represents significant differences. The details of the statistics in this figure are provided in Supplementary Data 1.

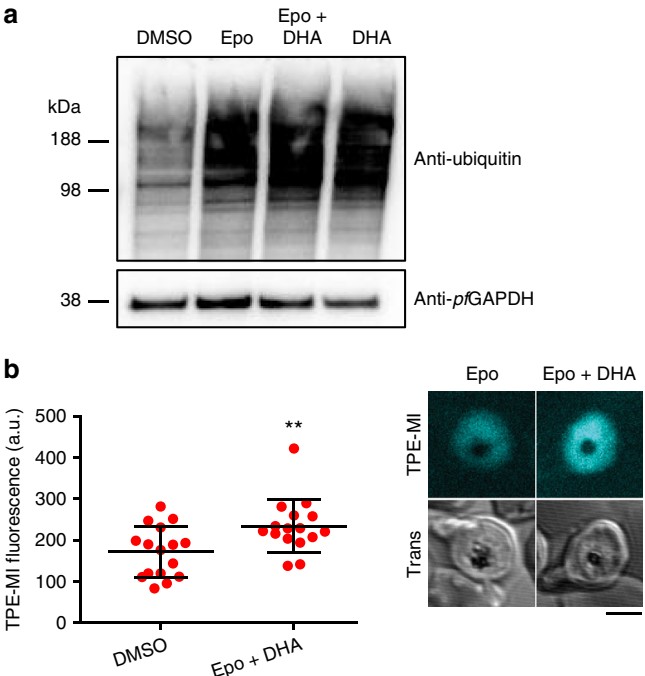

**Fig. 8** TPE reports on accumulation of unfolded proteins in malaria parasites (*P. falciparum*) treated with dihydroartemisinin (DHA) and epoxomicin. **a** Western blot of trophozoite stage *P. falciparum* treated with DHA and the proteasome inhibitor, epoxomicin (Epo) or DMSO control, for 3 h. **b** TPE-MI fluorescence values from individual trophozoites within red blood cells, measured by confocal microscopy. (*left*) Bars represent the mean ± s.d. *represents a significant difference. The details of the statistics in this figure are provided in Supplementary Data 1. (*right*) A representative set of images. *Scale bar*, 3 μm.

parasites by promiscuously damaging the proteome through free radical production, which leads to non-specific damage to proteins, resulting in a build-up of damaged and unfolded proteins and activation of stress responses[44–47]. We tested whether we could use TPE-MI as a probe of this imbalance in proteostasis. *Plasmodium falciparum* trophozoite-infected red blood cells were

treated with DHA in the presence of the proteasome inhibitor, epoxomicin (to suppress degradation of damaged proteins). This treatment led to an increase in total ubiquitinated proteins as previously described (Fig. 8a)[47]. DHA treatment led to significant elevations in TPE-MI reactivity over baseline conditions (Fig. 8b, c)—and also when compared to epoxomicin treatment alone (Supplementary Fig. 11). This result further supports the conclusion that DHA leads to a backlog of unfolded proteins as the basis for triggering the stress response.

## Discussion
Here we describe the use of the fluorigenic dye TPE-MI to probe for unfolded proteins by assaying the unmasking of cysteine thiols that are buried in the folded state of globular protein domains. This is an important step for understanding proteostasis mechanisms because unfolded proteins are highly permissive to misfolding and forming aggregates, which are a key end-product feature of neurodegenerative diseases[48, 49]. We validated the method with purified proteins and showed it can report on increases in unfolded protein load under diverse proteostatic stresses: heat shock, blockage of folding processing in the ER, hsp90 inhibition and free radical production.

Even though we expected TPE-MI to label surface-exposed cysteines which would contribute to the overall fluorescence yield, it was remarkable that we could detect a significant overall increase in fluorescence when we stressed proteome foldedness. This was especially apparent in the use of TPE-MI for proteomics-based analyses of unfolding. Our experiment was tailored for a simple assessment of the most abundant proteins in the proteome (by virtue of the 1-dimensional LC-fractionation step), but the method should be adaptable to broader (i.e., deeper) proteome coverage (e.g., by using 2-dimensional LC-based proteome fractionation methods), which offers the potential to systematically understand how the free cysteines change under proteome stresses. Furthermore, additional investigations of the TPE-MI concentration and time dependence of reactivity will likely capture a greater list of proteins of altered conformational status which could be most powerful for measuring the kinetic process of folding in live cells. The value of this capacity is highlighted by a recent study that used limited proteolysis on cell lysates to measure the thermodynamic stability of the proteome

upon chaotrope- and thermally mediated denaturation[50]. Our method here provides in principle a capacity to extend this approach by measuring folding kinetics of proteins that contain free cysteine thiols in live cells. Hence, more broadly, we believe TPE-MI offers the capacity to perform both simple and rapid tests of proteostasis health in essentially any cell-based format—as well as more elaborate assays of folding of individual proteins containing cysteine residues.

Application of TPE-MI to probe proteostasis imbalance in HD cell models yielded insight into the early imbalance events that precede the formation of visible aggregates. In the induced pluripotent stem cell (iPSC) -derived neural cell lines, imbalances were reflected in an attenuated capacity of cells to suppress accumulation of unfolded proteins upon interference of proteostasis with thapsigargin. Prior studies have indicated problems in polyQ-expanded knock-in Huntingtin cell lines in energy production, which is concordant with a broader problem in proteostasis[51]. Other work has also implicated proteostasis imbalance as crucial in Huntington Disease (reviewed in ref. [52]).

One particularly interesting finding from our studies was the dramatic loss of TPE-MI reactivity in cells with inclusions. Prior work has suggested that misfolded proteins (which arise from unfolded proteins) can form two classes of visible aggregates for processing misfolded proteins called the IPOD and JUNQ[39, 40]. While it is not clear whether the formation of IPOD or JUNQ structures alleviates proteostasis stress, prior studies have suggested that cells that form Httex1 inclusions, which partition into the IPOD structure, have a higher survival rate than cells retaining diffuse Httex1, when standardized to expression level[38]. Other evidence to support this finding is that soluble mutant Httex1 is actively recognised as abnormal and preferentially targeted for degradation compared to the wild-type Httex1 counterpart[53, 54]. In addition, it has been reported that ER stress pathways are activated by soluble pools of Httex1 before inclusion formation[55]. We recently found that when Httex1 forms inclusions, the cells enter a state of quiescence which deactivates the trigger of apoptosis from soluble Httex1 and switches cell death to a slower necrotic mechanism[56]. As inclusions matured there was a progressive coaggregation of prion-like proteins from the proteome[56]. Hence our findings raise the interesting possibility that soluble Httex1 strain proteostasis before inclusions are formed, which promotes the accumulation of unfolded proteins from the broader proteome. Thus the backlogged unfolded proteome may become co-aggregated en masse into the inclusion as quiescence is established and mask many of the exposed cysteine residues.

In summary, our new approach to measure unfolded load provides insight into proteostasis capacity in cells and opens pathways to study mechanisms of protein quality control. In addition, the approach provides possibilities for the development of biomarkers for early stage diagnosis of neurodegenerative and possibly other diseases.

## Methods

**Materials**. Tetrahydrofuran (THF, Labscan) was purified by distillation from sodium benzophenone ketyl under nitrogen prior to use. Maleic anhydride, sodium acetate, sodium carbonate, glutathione (GSH), guanidine hydrochloride (GuHCl), bovine β-lactoglobulin, yeast (Saccharomyces cerevisiae) enolase, bovine ubiquitin, NMM, tunicamycin and other reagents were purchased from Sigma-Aldrich and used as received. Human peroxiredoxin 3 was expressed and purified as described previously[57], and maintained in buffer with reducing agent tris(2-carboxyethyl) phosphine. Peroxiredoxin 3 was freshly desalted to remove reducing agents before use with a PD-10 column (GE Healthcare). Renilla luciferase vector was used from the Qiagen Cignal Heat Shock Response Reporter (luc) Kit. Luciferase activity was measure using Dual-Luciferase Reporter Assay System (Promega), as per the manufacturers protocol. The Httex1 fusion constructs were prepared as described previously[58]. All constructs were sequenced for verification.

**Synthesis of TPE-MI**. The precursor, 4-(1,2-diphenyl-2-(p-tolyl)vinyl)aniline (TPE-NH$_2$), was synthesised according to the previously reported procedure[16]. To a solution of TPE-NH$_2$ (0.361 g, 1.0 mmol) in dry THF (20 ml), maleic anhydride (0.196 g, 2.0 mmol) was added and stirred at room temperature for 12 h. The solvent of the resulting mixture was evaporated under reduced pressure and a yellow solid was obtained as an intermediate, which directly underwent further reaction without purification. Anhydrous sodium acetate (0.205 g, 2.5 mmol) was added into a solution of the intermediate in acetic anhydride (10 ml) and the mixture was stirred at 75 °C for 6 h. After cooling to ambient temperature, the reaction mixture was quenched with 20% Na$_2$CO$_3$ aqueous solution and extracted with dichloromethane three times. The organic layer was collected and washed by brine for three times. The crude product was concentrated and purified by silica-gel chromatography to furnish a light yellow solid (0.360 g). Yield = 82%. $^1$H NMR (400 MHz, CDCl$_3$): δ (tetramethylsilane (TMS), p.p.m.) 7.12–7.02 (m, 14 H), 6.93–6.90 (m, 4 H), 6.81 (d, J = 3.2 Hz, 2 H), 2.27 and 2.26 (two singlets, 3 H). $^{13}$C NMR (100 MHz, CDCl$_3$): δ (TMS, p.p.m.) 168.83, 143.13, 142.92, 142.76, 141.07, 139.90, 139.68, 138.74, 135.77, 135.59, 133.53, 131.31, 131.27, 130.78, 130.74, 130.70, 130.67, 130.55, 128.61, 127.95, 127.75, 127.16, 127.13, 127.07, 126.97, 126.01, 125.82, 20.59, 20.56. High-resolution mass spectrometry (HRMS) (matrix assisted laser desorption/ionization time-of-flight (MALDI-TOF), m/z): [M]$^+$ calcd. for C$_{31}$H$_{23}$NO$_2$, 441.1729; found, 441.1734.

**Synthesis of GSH-TPE-MI**. TPE-MI was synthesised and characterised as previously reported[16]. In essence, to a solution of TPE-MI (5 mg) in THF (4 ml), glutathione (GSH, reduced, 5 mg) in water was added dropwise with stirring. The reaction mixture was further stirred at room temperature for 12 h and the solvent was evaporated under reduced pressure. The final product was obtained after purification using high-performance liquid chromatography (HPLC) and dried under vacuum as white powders (6.7 mg). Yield = 80%. HPLC (λ = 330 nm): methanol as eluent, retention time 1.441 min. HRMS (ESI, m/z): [M-H]$^-$ calcd. for GSH-TPE-MI, 747.8359; found, 747.2462.

**Synthesis of non-alkylating TPE analogue**. The non-alkylating TPE analogue, 1-(p-bromophenyl)-2-(p-hydroxyethyl)-1,2-diphenylethylene was synthesised as described in ref. [59]. In essence, synthesis of 2-(4-(2-(4-bromophenyl)-1,2-diphenylvinyl)phenoxy)ethan-1-ol (1): 4-(2-Hydroxyethoxy)benzophenone (2) was synthesised according to previous literature[60]. To a solution of 2 (2 mmol, 484 mg), 4-bromobenzophenone (4 mmol, 1.04 g) and zinc powder (14.4 mmol, 941 mg) in dry THF, TiCl$_4$ (7.2 mmol, 0.79 ml) was added dropwise at −78 °C. Then the reaction mixture was warmed to ambient temperature and heated to reflux for 10–12 h. After cooling down, the reaction mixture was filtered and extracted with dichloromethane three times. The organic layer was combined, dried (MgSO$_4$) and concentrated under reduced pressure. The crude product was purified by silica gel chromatography with hexane/ethyl acetate (9:1) as eluent to afford the desired product (280 mg, 0.6 mmol) as pale yellow solid. Yield = 30%. $^1$H NMR (400 MHz, CDCl$_3$): δ (TMS, p.p.m.) 7.25–7.19 (m, 2 H), 7.14–7.09 (m, 6 H), 7.04–6.99 (m, 4 H), 6.95–6.86 (m, 4 H), 6.69–6.64 (m, 2 H), 4.05–3.99 (m, 2 H), 3.94–3.91 (m, 2 H). $^{13}$C NMR (100 MHz, CDCl$_3$): δ (TMS, ppm) 156.67, 156.57, 142.94, 142.84, 142.74, 142.33, 142.24, 140.40, 138.31, 138.28, 135.65, 135.54, 132.36, 131.93, 131.90, 130.66, 130.64, 130.29, 130.17, 127.24, 127.22, 127.11, 127.03, 126.98, 126.05, 125.95, 125.89, 125.63, 119.64, 113.20, 113.03, 76.71, 76.39, 76.08, 68.34, 68.30, 60.82. HRMS (MALDI-TOF), m/z calcd. for C$_{28}$H$_{23}$BrO$_2$: 470.0881; found 470.0845 (M$^+$).

**Instrument used for compound characterisation**. $^1$H and $^{13}$C NMR spectra were measured on a Bruker ARX 400 NMR spectrometer using chloroform-d as the deuterated solvent with tetramethylsilane (TMS; δ = 0) as the internal standard. Mass spectrum of non-alkylating TPE analogue was run by a Waters$^R$ Micromass$^R$ MALDI micro MX$^{TM}$ Mass Spectrometer operating on the reflectron mode with DCTB (trans-2-[3-(4-tert-Butylphenyl)-2-methyl-2-propenylidene]malononitrile) as matrix. Mass spectrum of GSH-TPE-MI was recorded on an Agilent Technologies 6520 Accurate-Mass Q-TOF LC/MS operating in an ESI negative ion mode. Ultraviolet–visible (UV–Vis) absorption spectra were measured on Cary 50 UV–Vis spectrometer. Steady-state fluorescence signals were recorded on a Cary Eclipse fluorimeter.

**In vitro protein and lysate unfolding assays**. Stock solutions of TPE-MI (1 mM) and proteins were prepared by dissolving appropriate amounts of the dye in dimethyl sulfoxide (DMSO) and the protein in PBS, respectively. In the unfolding experiments, protein (250 μM for β-lactoglobulin and ubiquitin; 50 μM for enolase; 5 μM for peroxiredoxin; 0.5 mg ml$^{-1}$ for the cell lysate) was incubated in 4.6 M GuHCl solution for 3 min before the addition of 50 μM TPE-MI into the mixture. Time-course fluorescence intensity was recorded by using the excitation and emission wavelengths of 350 and 470 nm, respectively, and time interval of 1 s. GSH competition experiment was performed by mixing an appropriate amount of protein and GSH in 4.6 M GuHCl for 3 min. Fluorescence intensity at 470 nm was recorded immediately after the addition of 50 μM TPE-MI into the mixture.

**Cell lysate preparation.** $7 \times 10^5$ HeLa cells were seeded in six-well plates. On the following day, cell were washed once in PBS and lysed by repeated pipetting in 300 μl native lysis buffer per well (20 mM Tris pH 8, 2 mM $MgCl_2$, 1% Triton X-100, 1 EDTA-free Complete protease inhibitor tablet (Roche)/10 ml, 150 μM NaCl, 20 Units $ml^{-1}$ Benzonase (EMDMillipore)). Protein concentrations were determined by bicinchoninic acid assay using bovine serum albumin as mass standard.

**Time-resolved fluorescence measurements.** Fluorescence decay measurements were performed by the time-correlated single-photon counting technique using the frequency doubled output (380 nm) of a mode-locked and cavity-dumped Titanium:sapphire laser (Coherent Mira 900f). The repetition rate of the laser pulses was reduced to 5.4 MHz (APE PulseSwitch). Emission was collected under magic angle conditions, spectrally selected (470 nm) using a Jobin Yvon H20 monochromator and detected with a microchannel plate photomultiplier (Eldy EM1-132-1). Data were acquired using a Becker and Hickl SPC-150 TCSPC card and SPCM software, and analysed using the FAST (Fluorescence Analysis Software Technology) software (Edinburgh Photonics).

**Mammalian cell line culturing and transfection.** Neuro-2a, HEK293 and HeLa cell lines (from lab cultures orginally obtained from ATCC) were used in this study and tested and cleared for mycoplasma. Cells were not tested for cross-contamination of other cell lines or misidentification. We justified their use in context of this in that the cell-line specific biology is not important to the general outcomes of the experiments. HeLa and HEK293 cells were maintained in Dulbecco's modified Eagles Medium (DMEM) supplemented with 1 mM glutamine, 100 Units $ml^{-1}$ penicillin-streptomycin and 10% v/v foetal bovine serum. Cells were cultured at 37 °C in a humidified incubator with 5% atmospheric $CO_2$. For microscopy experiments $7 \times 10^4$ HeLa cells were plated on eight-well μ-slides (Ibidi). For flow cytometry, $2.8 \times 10^5$ HeLa cells were plated on 12-well plates. Neuro-2a cells were maintained in OptiMem (Life Technologies) supplemented with 10% v/v foetal calf serum, 1 mM glutamine, 100 units per ml penicillin and 100 μg $ml^{-1}$ streptomycin in a humidified incubator with 5% atmospheric $CO_2$. For microscopy experiments $9 \times 10^4$ Neuro-2a cells were plated on 8-well μ-slides (Ibidi). For flow cytometry, $4 \times 10^5$ Neuro-2a cells were plated on 12-well plates. Cells were transfected using Lipofectamine 2000 according to the manufacturer's instructions with 0.5 μg DNA for the microscopy experiments and 1.6 μg DNA for the flow cytometry. Media was refreshed daily after transfection.

**hiPSC maintenance and differentiation.** CAG33 human iPSC (hiPSCs) (ND36997) and CAG180 iPSCs (ND36999) harbouring *HTT* alleles with 33 and 180 CAG repeats, respectively, were obtained from the NINDS iPSC Repository at Coriell Institute. The expanded *HTT* allele in the CAG180 hiPSC line was corrected to 18 CAG repeats by homologous recombination to generate isogenic control hiPSC lines: isoHD-Corr-1, isoHD-Corr-2, and isoHD-Corr-3[42]. All hiPSCs were cultured on Matrigel-coated plates in mTeSR-1 medium (STEMCELL Technologies, #05850) in a cell culture incubator at 37 °C and 5% $CO_2$.

hiPSCs were induced into pNSCs according to a previously published protocol[61]. Briefly, hiPSCs at about 20% confluence were treated with N2B27 media (DMEM-F12/Neural Basal medium 1:1 with 1% N2, 2% B27, 1% pen/strep/glutamine, 10 ng $ml^{-1}$ hLIF, and 5 μg $ml^{-1}$ BSA) containing 3 μM CHIR99021 (Tocris), 2 μM SB431542 (Sigma) and 0.1 μM compound E (Millipore) for the first 7 days. The culture was then split 1:3 for the next six passages using Accutase without compound E on Matrigel-coated plates. Cells at passage 5 were used for experiments.

**TPE-MI cell staining.** TPE-MI was dissolved in DMSO as 1 or 2 mM stocks. Stocks were kept at 4 °C in the dark. For experiments involving HeLa and Neuro2a cells, plated cells were rinsed with PBS and then treated with freshly diluted TPE-MI (50 μM in PBS) for 30 min at 37 °C. TPE-MI solution was removed. For flow cytometry, cells were resuspended in PBS, pelleted by centrifugation (120 *g* for 6 min) and then resuspended in 250 μl PBS in flow cytometry tubes. For imaging, cells were fixed on the plate with 4% (w/v) paraformaldehyde. For SDS–PAGE, the stained Neuro2a cells were lysed on the plate by addition of native lysis buffer and analysed by 12% acrylamide SDS–PAGE. Protein concentrations were determined by bicinchoninic acid assay with bovine serum albumin as mass standard. TPE-MI fluorescence was recorded on unstained gels with a Gel Imaging system (Bio-Rad). Then the gel was stained with Coomassie Blue for assay of total protein levels.

For experiments involving hiPSCs, pNSCs were treated with 1 μM thapsigargin (Sigma) or DMSO for 16 hours. Treated cells were rinsed once with PBS and then incubated with freshly diluted TPE-MI (50 μM in PBS) for 30 min at 37 °C. TPE-MI was removed and cells were detached using Accutase and filtered with 70 μM cell screener for flow cytometry analysis.

**Cytotoxicity test.** $7 \times 10^4$ HeLa cells were plated onto 12-well plates. For the dose curves, TPE-MI was applied for 30 min. For the time dependent experiment, 100 μM TPE-MI was applied for variable lengths of time. After the treatments, cells were detached in 400 μl PBS and assessed for viability using the Countess Cell counter (Invitrogen) with Trypan blue staining.

**NMM competition experiment.** $4 \times 10^5$ Neuro-2a cells were plated on 12-well plates. The next day cells were washed with 1 ml PBS and then with PBS supplemented with NMM for 30 min. The media was replaced with 50 μM TPE-MI for staining for 30 min. Cells were detached in 400 μl PBS and assessed for fluorescence in a Cary Eclipse fluorimeter.

**Tunicamycin and heat shock treatment.** Tunicamycin was added to plated HeLa and Neuro2a cells before TPE-MI treatment. For heat shock, plated HeLa cells were placed in a water bath at 42 °C for variable time periods as indicated. To measure protein unfolding of *Renilla* luciferase, cells expressing Qiagen Cignal Heat Shock Response Reporter (luc) plasmids were analysed for Renilla luminescence Dual-Luciferase Reporter Assay System (Promega), as per the manufacturers protocol. In this assay, *Renilla* luciferase was expressed via a constitutive CMV (cytomegalovirus) promoter and therefore changes in luminescence reflect loss of folding, rather than changes in expression. In the heat shock recovery experiments, media was changed after heat shock and cells returned to the 37 °C incubator.

**Proteomics sample preparation.** Neuro2a cells were cultured for 12 passages in SILAC-LYS6 (Catalogue Number: 89983, Thermo Fischer Scientific, San Jose, CA, USA) labelled media or unlabelled media. The SILAC-LYS6 labelled cells were treated with tunicamycin (1 μg $ml^{-1}$) in T25 flasks overnight, then both SILAC-LYS6 labelled and unlabelled cells were treated with TPE-MI (200 μM, 30 min). Cells were then detached using a cell scraper, lysed using RIPA buffer (50 mM Tris-HCl, 150 mM NaCl, Protease inhibitor, 1% v/v NP40, 2 mM EDTA and DNAase1), then centrifuged at 16,000 *g* for 10 min. The protein concentration in the supernatant was determined by a BCA assay (Catalogue number: 23225, Thermo Fischer Scientific) using BSA as the mass standard. Then, 50 μg of the total protein in each supernatant (i.e., tunicamycin-treated and control) (*n* = 5) was combined in a 1:1 ratio and extracted using methanol-chloroform precipitation[62]. The protein pellet was solubilized in 100 μl of 8 M urea, 50 mM triethylammonium bicarbonate, reduced using 10 mM tris(2-carboxyethyl)phosphine, pH 8.0, and alkylated with 10 mM iodoacetamide for 45 min, then subjected to trypsin digestion (2.5 μg, 24 °C, overnight). The resultant peptides were desalted by solid-phase extraction, by acidification in 1% v/v formic acid; pre-washing the cartridge (Oasis HLB 1 cc Vac Cartridge, product number 186000383, Waters Corp., USA) with 1 ml of 80% v/v acetonitrile (ACN) containing 0.1% v/v trifluoroacetic acid (TFA) and washing with 1.2 ml of 0.1% TFA three times by applying to a vacuum manifold; loading the sample on the cartridge and washing with 1.5 ml of 0.1% v/v TFA; eluting the sample with 0.8 ml of 80% v/v ACN containing 0.1% v/v TFA and collecting in 1.5 ml microcentrifuge tubes; then lyophilization by freeze drying (Virtis, SP Scientific). The peptides were resuspended in 100 μl distilled water. A microBCA assay (Catalogue Number: 23235, Thermo Fischer Scientific) with BSA as the mass standard was performed in triplicate to quantify the peptides. Then, a 2 μg aliquot of peptides was dried and resuspended in 15 μl of 2% v/v ACN containing 0.05% v/v trifluoroacetic acid before analysis.

**NanoESI-LC-MS/MS analysis.** Samples were analysed by nanoESI-LC-MS/MS using a Orbitrap Fusion Lumos mass spectrometer (Thermo Scientific) fitted with a nanoflow reversed-phase-HPLC (Ultimate 3000 RSLC, Dionex). The nano-LC system was equipped with an Acclaim Pepmap nano-trap column (Dionex—C18, 100 Å, 75 μm × 2 cm) and an Acclaim Pepmap RSLC analytical column (Dionex—C18, 100 Å, 75 μm × 50 cm). For each LC-MS/MS experiment, 1 μl (0.135 μg protein) of the peptide mix was loaded onto the enrichment (trap) column at an isocratic flow of 5 μl $min^{-1}$ of 3% $CH_3CN$ containing 0.1% v/v formic acid for 5 min before the enrichment column was switched in-line with the analytical column. The eluents used for the LC were 0.1% v/v formic acid (solvent A) and 100% $CH_3CN$/0.1% formic acid v/v (solvent B). The gradient used (300 nl $min^{-1}$) was from 3–22% B in 90 min, 22–40% B in 10 min and 40–80% B in 5 min then maintained for 5 min before re-equilibration for 8 min at 3% B prior to the next analysis. All spectra were acquired in positive mode with full scan MS from *m/z* 400–1500 in the FT mode at 120,000 mass resolving power after accumulating to a target value 5.00e5 with maximum accumulation time of 50 ms. Lockmass of 445.12002 was used. Data-dependent HCD MS/MS of charge states > 1 was performed using a 3 s scan method, at a target value of 1.00e4, a maximum accumulation time of 60 ms, a normalised collision energy of 35%, an activation Q of 0.25, and at a 15,000 mass resolving power. Dynamic exclusion was used for 45 s.

**Proteomic data analysis.** Data analysis was carried out using Proteome Discoverer (version 2.1.0.81; Thermo Scientific) with the Mascot search engine (Matrix Science version 2.4.1) against the Swissprot *Mus Musculus* database (Version: 2015_07; 548,872 entries). The search was conducted with 20 p.p.m. MS tolerance, 0.6 Da MS/MS tolerance, with one missed cleavage allowed. The following modifications were allowed: methionine oxidation, N-terminal protein acetylation, N-terminal methionine cleavage and SILAC-LYS6 (variable); carbamidomethylcysteine (fixed). The false discovery rate maximum was set to 0.005% at the peptide identification level (actual was 0.005 for each replicate) and 1% at the protein identification level. Proteins were filtered for those containing at least one unique peptide in all five replicates. The common contaminant, Keratin, was excluded from the data set. Peptide quantitation was performed in Proteome

Discoverer v. 2.1.0.81 using the precursor ion quantifier node with the mass tolerance for isotope pattern multiplets set to 2 p.p.m., and two single peak/missing channels allowed. The protein abundance in each replicate was calculated by summation of the unique peptide abundances that were used for quantitation (heavy SILAC-LYS6 and light unlabelled derivatives). Quantified proteins were then filtered to those containing at least two unique peptides, one of which contained cysteine. The average peptide abundance ratio (i.e., tunicamycin-treated versus control) for the non-cysteine-containing peptides were then calculated for each protein. These values were then compared with the peptide abundance ratio of the corresponding cysteine-containing peptide(s) for each protein, in order to quantify the change in cysteine peptide abundance following TPE-MI treatment. Wilcoxon matched pair sign rank test was performed to calculate the statistical significance (Graphpad Prism, v.7) of cysteine depletion on TPE-MI treatment.

**Malaria parasite experiments**. Early trophozoite stage *P. falciparum*-infected red blood cells (21–24 h post invasion, 5% haematocrit) were incubated with 1 μM DHA and 400 nM epoxomicin or with 400 nM epoxomicin alone in the presence of 0.1% DMSO (carrier solvent), for 3 h at 37 °C. Treated parasite cultures were attached to erythroagglutinin PHA-E-coated glass slides, incubated in 50 μM TPE-MI for 30 min at 37 °C and fixed with 2% formaldehyde and 0.008% glutaraldehyde before imaging.

For the western blot experiment, red blood cells infected with trophozoite stage *P. falciparum* at 5% parasitemia and 5% haematocrit, were exposed to vehicle (0.1% v/v DMSO) or 400 nM epoxomicin and/or 1 μM DHA for 3 h. Parasites were harvested for analysis of ubiquitinated proteins as previously described[47]. The antibodies used were rabbit anti-ubiquitin (Dako-Z0458) and goat anti-rabbit IgG-peroxidase (Sigma-Aldrich-A0545), with dilutions 1:100 and 1:25,000, respectively.

**Flow cytometry**. Cells were analysed at a high flow rate in an LSRFortessa flow cytometer (BD Biosciences). 100,000–200,000 events were collected using a forward scatter threshold of 5000. Data were collected in pulse height, area and width parameters for each channel. For TPE-MI, data were collected with the 355 nm laser and a $450 \pm 50$ nm bandpass filter. For mCherry, data were collected with the 561 nm laser and $621 \pm 20$ nm bandpass filter. Flow cytometry data were analysed with FlowJo (Tree Star Inc.) and graphs were analysed in Prism 5. For PulSA, we analysed the data as described previously[41, 54], and used the non-aggregating protein samples to set the boundary for the ni gate. The gating strategies are shown in Supplementary Fig. 12a–f including TPE-MI signal gate.

For hiPSC experiments, cells were analysed using a BD FACSAria flow cytometer and 10,000 events were collected with a 355 nm laser. Cell parameters including pulse height, area and width were used for the analysis.

**Imaging**. After staining with TPE-MI, cells were fixed with 4% (v/v) paraformaldehyde in PBS for 15 min. For the cells treated with tunicamycin or heat shock, or examined for mCherry fluorescence, cells were imaged on a Zeiss LSM780 Confocal microscope for TPE-MI (excitation: 355 nm, emission: 445–500 nm) and mCherry (excitation: 561 nm, emission: 600–700 nm) using a ×40 objective lens. Cells that tranfected with Httex1 were imaged 48 h after transfection.

For image quantitation, mean TPE-MI intensity within regions of interest were calculated by Fiji (ImageJ) in the cytosol of mammalian cells and within the plasmodium inside red blood cells of blood cells only containing a single parasite.

The mean TPE-MI signal, and corresponding Coomassie Blue pixel intensities, in SDS–PAGE images were measured on region of interests encompassing the entire lane for each sample.

**Statistics**. Statistical analyses were performed using Prism (Graphpad) software packages. The exact *P* values, raw values and statistical details are provided in Supplementary Data 1.

**Data availability**. The mass spectrometry proteomics data have been deposited to the ProteomeXchange Consortium via the PRIDE[63] partner repository with the data set identifier PXD006527 and 10.6019/PXD006527. All other data are available from the corresponding author on reasonable request.

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

## Acknowledgements

We thank Peng Zeng and Sijie Chen from The University of Melbourne for technical support and helpful discussion. We also thank Michael Parker from St Vincents Institute of Medical Research, VIC, Australia for helpful discussion and feedback. We thank Juliet Gerrard from the University of Auckland, NZ for provision of the human peroxiredoxin expression construct. This work was supported by grants to D.M.H (National Health and Medical Research Council grants APP1049458, APP1049459 and APP1102059 and Australian Research Council (FT120100039 and DP170103093)) Y.H.(Australian Research Council DE170100058, Bruce Stone Fellowship from La Trobe University, Melbourne Neuroscience Institute Interdisciplinary Seed Funding (University of Melbourne), and McKenzie Fellowship from University of Melbourne) and grants to M.A.P. (Ministry of Education Singapore Tier 1 grant R-172-000-297-112, and Agency for Science, Technology and Research Strategic Positioning Fund for Genetic Orphan Diseases grant SPF2012/005).

## Author contributions

D.M.H. and Y.H. conceived the ideas and were responsible for the overall project management. D.M.H., Y.H., L.T., M.A.P., J.B., G.E.R. and Z.C. designed the research. Z.C., N.S.M, M.R., R.J.W., J.B., T.A.S., Z.S., B.Z.T., Y.H., X.X., L. T., M.A.P. and D.M.H. performed experiments and/or interpreted the data. Z.C., Y.H. and D.M.H. wrote the manuscript with feedback from the other authors.

## Additional information

**Competing interests:** The authors declare no competing financial interests.

