## [Peer Review File · Nature Communications]

Reviewers' Comments:

Reviewer #1 (Remarks to the Author):

The manuscript by Chen et. al " A thiol probe for measuring unfolded protein load and proteostasis in cells" describes a novel probe (tetraphenylethene conjugated to maleimide, TPE-MI) to monitor the folded state of proteins in live cells. TPE-MI fluoresces upon labeling free and exposed cystein residues in vitro, and the authors argue that it can be used to study global misfolding events in live cells. They focus on a polyQ Htt model as a negative modulator of proteostasis.

Although multiple tools exist for monitoring misfolding in lysate, a live cell tool for visualizing misfolding in live cells would be tremendously useful.

One serious drawback of the TPE-MI probe, in my opinion, is that it can not be used at the single cell level (because the signal to noise is just too low). As such its usefulness is severely limited due to the inherent variability of proteome-wide events across a population of cells, coupled with the already high signal to noise. Moreover, even at the population, level, the reported differences are quite small.

In order to demonstrate the usefulness of the live cell approach with TPE-MI, the study should examine a relevant global phenomenon, such as (for example) a decrease in proteostasis over the course of aging, with multiple data points and easy controls. The polyQ experiment is insufficient because it is very noisy and relies on many assumptions about what we think is happening there. What about using inhibitors of different chaperones, like Hsp70 and 90, or proteasome inhibitors? The TPE-MI results would then be correlated with a biochemical assay for unfolding/aggregation. I think that these experiments might improve the study, beyond the small effect observed with tunicamycin.

Reviewer #2 (Remarks to the Author):

This is an interesting and wide-ranging paper that describes a potentially novel probe for unfolded proteins. The data on the initial characterisation of the probe are reasonably convincing and the authors have used the probe to report on both "unfolded" huntingtin and malarial parasites.

However, I am not convinced, from the experiments presented, that the specificity of the probe for unfolded proteins has been demonstrated.

My main concern is that in Fig 4C, a very strong band appears at ~55kDa in the tunicamycin treated samples. This suggests that the probe has a preference for binding a limited group of stress-responsive proteins, perhaps one of the PDI family, as PDIs are centrally involved in thiol-disulfide exchange reactions in the ER. I would like to see the authors identify this protein by mass spec and then show that knock-down of the gene in e.g. Neuro2a cells does not change the ability of TPE-MI to report on the overall unfolded protein load of cells after tunicamycin stress.

In Fig 4A, a co-stain with known ER/cytoplasmic markers is required to fully understand how the TPE-MI is distributed within cells.

Although the authors have used a ubiquitin control in Fig 2, further specific controls are required, namely B-lactoglobulin, enolase and/or peroxiredoxin 3 mutants lacking the core free cysteine.

In Fig 3/S3, the authors show data that TPE-MI is non-fluorescent in the presence of GSH and contrast this to previous results. Some of these experiments were performed in the presence of 4.6M GuHCl and DMSO carrier. The authors should show how the concentration of GuHCl influences the fluorescence of TPE-MI in the presence of GSH.

In Fig 7, there is a large spread of TPE-MI fluorescence in the DHA treated parasites. Although the result appears significant, I feel that more data is required to understand why there is a large variation in the dataset; for example, a control for free radical production would show whether or not variation in DHA fluorescence is directly related to variation in free radicals. An independent measure of unfolded protein content (e.g. ubiquitinated protein load after proteasome inhibitor treatment) is also needed here as a positive control.

Unfolded proteins are not the same as misfolded proteins, and this point could be clarified further in the text with respect to huntingtin and potential applications in other neurological diseases.

Reviewers' comments:

Reviewer #1 (Remarks to the Author):

The manuscript by Chen et. al " A thiol probe for measuring unfolded protein load and proteostasis in cells" describes a novel probe (tetraphenylethene conjugated to maleimide, TPE-MI) to monitor the folded state of proteins in live cells. TPE-MI fluoresces upon labeling free and exposed cystein residues in vitro, and the authors argue that it can be used to study global misfolding events in live cells. They focus on a polyQ Htt model as a negative modulator of proteostasis.

Although multiple tools exist for monitoring misfolding in lysate, a live cell tool for visualizing misfolding in live cells would be tremendously useful.

One serious drawback of the TPE-MI probe, in my opinion, is that it can not be used at the single cell level (because the signal to noise is just too low). As such its usefulness is severely limited due to the inherent variability of proteome-wide events across a population of cells, coupled with the already high signal to noise. Moreover, even at the population, level, the reported differences are quite small.

Q1. We appreciate the reviewer's general point here in that the analysis of single cells would be most useful. However, as with almost all assays in biology and biochemistry, a capacity to analyze single cells is not a death knell for its usefulness when the assay can robustly compare populations of cells. In this regard, our new strategy to directly detect unfolded protein load is a tremendous leap in providing a new capability that is not otherwise established in the literature. Our new proteomics data (as per Q6) shows how the method can be used to extract information about the biology of protein folding changes and we think illustrates superbly the power of what can be done at the population level.

It is also very important to clarify that "signal to noise", by its definition, is not a problem with the method. Rather there is a high baseline level of fluorescence as anticipated from thiol reactivity in folded and possibly intrinsically disordered proteins, which means that several cells are needed to get statistically significant results (as we have done in the work – including the analysis of groups of single cells from images – the plasmodium data). As such, this is not inherently a problem in the use of this system to measure changes in unfolded protein load.

In order to demonstrate the usefulness of the live cell approach with TPE-MI, the study should examine a relevant global phenomenon, such as (for example) a decrease in proteostasis over the course of aging, with multiple data points and easy controls.

Q2. It is important to clarify that our manuscript has already applied a "global" phenomenon of heat shock, which involved the collection of multiple data points tracking the recovery of proteostasis. We agree that there is intense interest in how proteostasis changes upon ageing. However, we disagree

with the reviewer that measuring ageing (such as in “old” cells or animal models) would be straightforward to perform in a controlled manner. This would be an ambitious project in its own right and more suitable to a follow up study.

The polyQ experiment is insufficient because it is very noisy and relies on many assumptions about what we think is happening there.

Q3. We are unsure of the exact concerns reviewer is referring to in this experiment – we have published the flow cytometry method previously (Ramdzan et al Nature Methods 2012) and the data holds up to statistical scrutiny as reported in the manuscript.

What about using inhibitors of different chaperones, like Hsp70 and 90, or proteasome inhibitors?

Q4. We have now applied the method to other proteome stresses. This includes novobiocin, which inhibits Hsp90 (without activating the heat shock response) and hydrogen peroxide. Both treatments significantly increased TPE reactivity in accordance with their anticipated influences on increasing levels of unfolded or misfolded proteins. This data has been added to the manuscript in Fig 4.

The TPE-MI results would then be correlated with a biochemical assay for unfolding/aggregation. I think that these experiments might improve the study, beyond the small effect observed with tunicamycin.

Q5. As stated above, we have already included heat shock as another stress to support the results seen with tunicamycin. Our revised manuscript now also includes additional stresses from Hsp90 inhibition and hydrogen peroxide as per Q4. Our new proteomics data also significantly extends the insight of the tunicamycin treatment and provides exciting new leads to biological mechanisms (see Q6)

Reviewer #2 (Remarks to the Author):

This is an interesting and wide-ranging paper that describes a potentially novel probe for unfolded proteins. The data on the initial characterisation of the probe are reasonably convincing and the authors have used the probe to report on both “unfolded” huntingtin and malarial parasites.

However, I am not convinced, from the experiments presented, that the specificity of the probe for unfolded proteins has been demonstrated.

My main concern is that in Fig 4C, a very strong band appears at ~55kDa in the tunicamycin treated samples. This suggest that the probe has a preference for binding a limited group of stress-responsive proteins, perhaps one of the PDI family, as PDIs are centrally involved in thiol-disulfide exchange reactions in the ER. I would like to see the authors identify this protein by mass spec and then show that knock-down of the gene in e.g. Neuro2a cells does not change the ability of TPE-MI to report on the overall unfolded protein load of cells after tunicamycin stress.

Q6. We have now performed a quantitative proteomics experiment to identify exactly which proteins are preferentially labeled with TPE under tunicamycin versus a control treatment. Our approach was to use SILAC to compare the abundance of peptides from control versus tunicamycin and how cysteine-containing peptides were selectively “lost” as a result of TPE conjugation, relative to non-cysteine peptides for each identified protein. This treatment showed that tunicamycin significantly increased reactivity of cysteines across the proteome by on average 6% (P= 0.0007 Wilcoxon signed-rank test). (Note that this number is not directly relatable to absolute extents of changes in Cys reactivity as measured in the fluorescence assays since the ratios are not weighted to the abundance of the different peptides). What this experiment does provide is fascinating insight into the considerable variation at the individual protein level. Several proteins showed many fold changes in reactivity to cysteine (top 6 hits were THRB, G3P, FHL1, ACTG, FBRL and ENOB). None of these proteins have reported functions in redox biology and there is no apparent functional connection between these proteins by network analysis (STRING). Of these proteins, ENOB as the closest mass (47 kDa) and is predicted to be a very abundant protein (top 5% based on the PaxDb: Protein Abundance Database (<http://pax-db.org/protein/2105743/Eno3>)).

This new data is a major new addition to the manuscript (Fig 6 and Supplementary Fig 9) and provides far more depth to understanding the changes in Cys-reactivity across the proteome.

In Fig 4A, a co-stain with known ER/cytoplasmic markers is required to fully understand how the TPE-MI is distributed within cells.

Q7. We have now performed the suggested experiment and placed it in Fig 4.

Although the authors have used a ubiquitin control in Fig 2, further specific controls are required, namely B-lactoglobulin, enolase and/or peroxiredoxin 3 mutants lacking the core free cysteine.

Q8. The experiments suggested by the reviewer are not trivial to do since it would require mutagenesis that is likely to destabilize the proteins and also requires the development of new expression and purification schemes (since some of these proteins were isolated from endogenous sources). As an alternative approach to respond to the reviewers query, we further tested the importance of cysteine reactivity in TPE-MI fluorescence by repeating the experiment with beta-lactoglobulin in the presence of NMM, which competes with TPE for thiol reactivity. There was no induction of fluorescence under this condition. This data is now added as a panel to figure 2.

In Fig 3/S3, the authors show data that TPE-MI is non-fluorescent in the presence of GSH and contrast this to previous results. Some of these experiments were performed in the presence of 4.6M GuHCl and DMSO carrier. The authors should show how the concentration of GuHCl influences the fluorescence of TPE-MI in the presence of GSH. I believe this has been done right?

Q9. Indeed we had already done these experiments in GuHCL and had missed labeling the data fully in the figure legend. We thank the reviewer for alerting us to this oversight. We have amended the figure legend accordingly.

In Fig 7, there is a large spread of TPE-MI fluorescence in the DHA treated parasites. Although the result appears significant, I feel that more data is required to understand why there is a large variation in the dataset; for a example, a control for free radical production would show whether or not variation in DHA fluorescence is directly related to variation in free radicals. An independent measure of unfolded protein content (e.g. ubiquitinated protein load after proteasome inhibitor treatment) is also needed here as a positive control.

Q10. We have now included new data (in Fig 5) that shows that free radical production can stimulate an increase in TPE reactivity. We have also performed an experiment to assess the level of damaged proteins by Western Blot in malaria after DHA treatment probing for ubiquitin and included this data in the figure. With respect to determining the reason for the variation, this is likely to be a technical issue since plasmodium are imaged inside host blood cells, which provide an additional parameter of variation. Furthermore the plasmodium are very small compared to mammalian cells which makes them generally more difficult to image by microscopy.

Unfolded proteins are not the same as misfolded proteins, and this point could be clarified further in the text with respect to huntingtin and potential applications in other neurological diseases.

Q11. We have revised the text to include this clarification.

Reviewers' Comments:

Reviewer #1 (Remarks to the Author):

The authors have improved their manuscript. TPE-MI will be a useful tool for the community, and therefore I support publication at this stage.

Reviewer #2 (Remarks to the Author):

The authors have added substantive new data to their paper to address my initial concerns, notably using proteomics to characterise the major proteins that TPE reacts with.

Additionally, a number of requested controls have been included for the data shown in Figures 2, 4 and 5.

I have no further concerns or comments for the authors.

Reviewer #3 (Remarks to the Author):

This manuscript describes that TPE-MI, a novel probe that is able to conjugate with free cysteine thiols of protein via maleimide reaction and is readily transformed into a fluorogen showing an aggregation-induced emission (AIE) property, can measure both in vitro unfolded protein load and in vivo proteostasis in cells. Although this study provided comprehensive evidence attempting to establish such a claim, the readout of fluorescence intensity was very low for in vivo conditions when observing conjugation of the free thiol with TPE-MI, so it is harder to predict that this probe will likely have broader applications. Several additional issues regarding previous reviewers' comments as well as the insights from some results (both original and revised data) should be clarified.

Major critiques

Q1. In Figure 2 from the in vitro study, the plots of three example proteins, bovine β -lactoglobulin, yeast enolase, and human Peroxiredoxin-3 [1/2Vmax at Kt1/2: ~375 at 1500 (s); ~75 at <500 (s); and ~90 at 750 (s), respectively] display significant difference in the curve patterns of TPE (as well as buffer) and its fluorescence intensity over time, indicating that TPE binding affinity and selectivity to these proteins vary. Therefore, TPE as a thiol reactive probe for in vivo cell-based studies, its binding affinity and selectivity should be addressed, esp. related to the claim from Figure 6 and Supplemental Figures.

Q2. In addition, in Figure 4 from these in vivo studies, the TPE fluorescence intensity (a.u.) in live HeLa cells, however, under various stress conditions, dropped significantly (>30~140 fold) to 3~5 (a.u.) compared to the in vitro fluorescence intensity (max 90~700, a.u.). Similar results with a small difference in TPE fluorescence intensity indicated in Figure 7A and 7B under extreme conditions by "comparing the wild-type polyQ length (25Q) with a severe mutant

(97Q)", but less severe with only as few as 40Q. These results revealed that even though showing statistical significance, the readout of TPE-MI barely differentiates such small net changes in cell-based studies, thus suggesting it is less applicable to other pathophysiological conditions with much smaller differentiation under in vivo conditions of proteostasis.

Q3. Furthermore, consistent with the critiques from other reviewers, results in Figure 5 suggest TPE selectively labels certain types of proteins (such as MW ~50) over others. Whether these identified ones for their TPE-labeled cysteine residues should be carefully examined in order to make claims that these cysteines are buried in the folded state of globular protein domains.

Minor request:

Q1. The manuscript didn't describe the specific setting for each TPE-MI fluorescence readout across all the results in order to obtain the "TPE fluorescence (a.u.)". Author may include the specific setting parameters for TPE-MI fluorescence readout.

In summary on the major claims of the paper, this thiol reactive dye can be used for in vitro conditions, but may face challenges for in vivo studies, as well as for broader applications. Since in vivo data were subtle, it may not be easy to overcome under the present structure (TPE-MI) or approach without enhancement of the fluorescence signal/noise ratios. Expectedly, this probe may partially be of interest to others in the community and the wider field.

REVIEWERS' COMMENTS:

Reviewer #1 (Remarks to the Author):

The authors have improved their manuscript. TPE-MI will be a useful tool for the community, and therefore I support publication at this stage.

There is no issue to address

Reviewer #2 (Remarks to the Author):

The authors have added substantive new data to their paper to address my initial concerns, notably using proteomics to characterise the major proteins that TPE reacts with.

Additionally, a number of requested controls have been included for the data shown in Figures 2, 4 and 5.

I have no further concerns or comments for the authors.

There is no issue to address

Reviewer #3 (Remarks to the Author):

This manuscript describes that TPE-MI, a novel probe that is able to conjugate with free cysteine thiols of protein via maleimide reaction and is readily transformed into a fluorogen showing an aggregation-induced emission (AIE) property, can measure both in vitro unfolded protein load and in vivo proteostasis in cells. Although this study provided comprehensive evidence attempting to establish such a claim, the readout of fluorescence intensity was very low for in vivo conditions when observing conjugation of the free thiol with TPE-MI, so it is harder to predict that this probe will likely have broader applications. Several additional issues regarding previous reviewers' comments as well as the insights from some results (both original and revised data) should be clarified.

Major critiques

Q1. In Figure 2 from the in vitro study, the plots of three example proteins, bovine β -lactoglobulin, yeast enolase, and human Peroxiredoxin-3 [1/2Vmax at Kt1/2: ~375 at 1500 (s); ~75 at <500 (s); and ~90 at 750 (s), respectively] display significant difference in the curve patterns of TPE (as well as buffer) and its fluorescence intensity over time, indicating that TPE binding affinity and selectivity to these proteins vary. Therefore, TPE as a thiol reactive probe for in vivo cell-based studies, its binding affinity and selectivity should be addressed, esp. related to the claim from Figure 6 and Supplemental Figures.

Q1. The reviewer raises a point we agree warrants further investigation in the future but we believe it is beyond the scope of the current study. This is because we performed the experiment in Fig 6 specifically to address a previous request from a reviewer to determine which proteins are being differentially labelled under tunicamycin stress. To extend this further again as suggested by the reviewer would not only involve a substantial further line of experiments (eg a time and concentration dependent investigation of TPE reactivity to the proteome by proteomics) but is also somewhat tangential (as a downstream application) to the key message of this study. This would be

a terrific experiment for a follow up investigation to get the kinetic information of rates and will likely unearth a much richer set of data as to how proteins fold. We have added a note of this point to the discussion section.

Q2. In addition, in Figure 4 from these in vivo studies, the TPE fluorescence intensity (a.u.) in live HeLa cells, however, under various stress conditions, dropped significantly (>30~140 fold) to 3~5 (a.u.) compared to the in vitro fluorescence intensity (max 90~700, a.u.). Similar results with a small difference in TPE fluorescence intensity indicated in Figure 7A and 7B under extreme conditions by “comparing the wild-type polyQ length (25Q) with a severe mutant (97Q)”, but less severe with only as few as 40Q. These results revealed that even though showing statistical significance, the readout of TPE-MI barely differentiates such small net changes in cell-based studies, thus suggesting it is less applicable to other pathophysiological conditions with much smaller differentiation under in vivo conditions of proteostasis.

Q2. The absolute fluorescence values are not on the same scale between the different experiments (ie the different graphs) because the fluorescence was measured under different conditions (and in some cases different calibration settings). Thus the signals cannot be directly compared as the reviewer has done. The magnitude of the differences from the baseline position provides a better comparable measure of the differences in fluorescence. We have amended the text in the Figure 2 legend to make this point clearer.

Q3. Furthermore, consistent with the critiques from other reviewers, results in Figure 5 suggest TPE selectively labels certain types of proteins (such as MW ~50) over others. Whether these identified ones for their TPE-labeled cysteine residues should be carefully examined in order to make claims that these cysteines are buried in the folded state of globular protein domains.

Q3. As per Q1, this experiment was initially performed to address queries from prior reviewers comments to determine which proteins were being labelled by the tunicamycin treatment. Hence, while we appreciate the reviewers comment, we feel it best to draw a line in the sand on this point as it increasingly drifts from the key message of the study and requires a substantial line of additional experiments. Further analysis of these proteins would make a great follow-up investigation.

Minor request:

Q1. The manuscript didn't describe the specific setting for each TPE-MI fluorescence readout across all the results in order to obtain the “TPE fluorescence (a.u.)”. Author may include the specific setting parameters for TPE-MI fluorescence readout.

Q4. The settings were in fact different and unfortunately we didn't keep a complete record of the entire fluorescence settings. Nonetheless, fluorescence units are typically an arbitrary scale and we believe that this is now clearer with the changes we made as per Q2.

In summary on the major claims of the paper, this thiol reactive dye can be used for in vitro conditions, but may face challenges for in vivo studies, as well as for broader applications. Since in vivo data were subtle, it may not be easy to overcome under the present structure (TPE-MI) or approach without enhancement of the fluorescence signal/noise ratios. Expectedly, this probe may partially be of interest to others in the community and the wider field.

Q4. For reasons we have mentioned in previous revisions, we disagree with the reviewer. In essence, the “signal to noise”, by its definition, is not a problem with the method and is a property of

the biology rather than probe. There is a high baseline level of fluorescence as anticipated from thiol reactivity in folded and possibly intrinsically disordered proteins. Differences in these levels attributable to proteome folding stresses are detectable statistically. As such, this is not inherently a problem in the use of this system to measure changes in unfolded protein load. This is in fact illustrated by our proteomics work and we believe there is a significant potential for the method to be used to probe the kinetic process of proteome folding in the future. A recent paper in *Science*¹ has illustrated interest in understanding proteome thermodynamic stability and we believe our method offers potential to take this to the next step by measuring folding kinetics in live cells. The approach by Leuenberger is restricted to assay of cell lysates which limits its potential to measure kinetics. We have now added an additional sentence to the discussion to elaborate on this point.

Reference:

1. Leuenberger, P. et al. Cell-wide analysis of protein thermal unfolding reveals determinants of thermostability. *Science* **355**(2017).